# Dermal Fibroblasts Internalize Phosphatidylserine-Exposed Secretory Melanosome Clusters and Apoptotic Melanocytes

**DOI:** 10.3390/ijms21165789

**Published:** 2020-08-12

**Authors:** Hideya Ando, Satoshi Yoshimoto, Moemi Yoshida, Nene Shimoda, Ryosuke Tadokoro, Haruka Kohda, Mami Ishikawa, Takahito Nishikata, Bunpei Katayama, Toshiyuki Ozawa, Daisuke Tsuruta, Ken-ichi Mizutani, Masayuki Yagi, Masamitsu Ichihashi

**Affiliations:** 1Department of Applied Chemistry and Biotechnology, Okayama University of Science, Okayama 700-0005, Japan; t17sd03ys@ous.jp (S.Y.); moemi@mukogawa-u.ac.jp (M.Y.); t20am02sn@ous.jp (N.S.); ryo-tado@dac.ous.ac.jp (R.T.); 2Frontiers of Innovative Research in Science and Technology (FIRST), Konan University, Kobe 650-0047, Japan; haruka-kohda@dspr.co.jp (H.K.); ishikawa-mam@awi.co.jp (M.I.); nisikata@konan-u.ac.jp (T.N.); 3Department of Dermatology, Osaka City University Graduate School of Medicine, Osaka 545-8585, Japan; chelsea.stamfordbridge.theblue@gmail.com (B.K.); ozawa@med.osaka-cu.ac.jp (T.O.); dtsuruta@med.osaka-cu.ac.jp (D.T.); 4Laboratory of Stem Cell Biology, Graduate School of Pharmaceutical Sciences, Kobe Gakuin University, Kobe 650-8586, Japan; mizutani@pharm.kobegakuin.ac.jp (K.M.); mm_ichihashi@hotmail.com (M.I.); 5Rosette Co., Tokyo 140-0004, Japan; myagi@rosette.co.jp; 6Anti-Aging Medical Research Center, Doshisha University, Kyoto 610-0394, Japan; 7Arts Ginza Clinic, Tokyo 105-0004, Japan

**Keywords:** apoptosis, dermis, epidermis, fibroblast, macrophage, melanin, melanocyte, melanosome, phosphatidylserine, pigmentation

## Abstract

Pigmentation in the dermis is known to be caused by melanophages, defined as melanosome-laden macrophages. In this study, we show that dermal fibroblasts also have an ability to uptake melanosomes and apoptotic melanocytes. We have previously demonstrated that normal human melanocytes constantly secrete melanosome clusters from various sites of their dendrites. After adding secreted melanosome clusters collected from the culture medium of melanocytes, time-lapse imaging showed that fibroblasts actively attached to the secreted melanosome clusters and incorporated them. Annexin V staining revealed that phosphatidylserine (PtdSer), which is known as an ‘eat-me’ signal that triggers the internalization of apoptotic cells by macrophages, is exposed on the surface of secreted melanosome clusters. Dermal fibroblasts were able to uptake secreted melanosome clusters as did macrophages, and those fibroblasts express TIM4, a receptor for PtdSer-mediated endocytosis. Further, co-cultures of fibroblasts and melanocytes demonstrated that dermal fibroblasts internalize PtdSer-exposed apoptotic melanocytes. These results suggest that not only macrophages, but also dermal fibroblasts contribute to the collection of potentially toxic substances in the dermis, such as secreted melanosome clusters and apoptotic melanocytes, that have been occasionally observed to drop down into the dermis from the epidermis.

## 1. Introduction

Melanosomes are specialized organelles produced within melanocytes in which melanin pigment is synthesized after which they are ultimately transferred to different types of cells. In the epidermis, melanosomes are transferred from melanocytes to neighboring keratinocytes to protect keratinocytes against solar ultraviolet light-induced nuclear DNA damage. Four possible mechanisms of melanosome transfer between melanocytes and keratinocytes have been proposed, that is, the cytophagocytosis model, the membrane fusion model, the shedding-phagocytosis model and the exocytosis-endocytosis model [1]. In 3 of those models, phagocytosis plays a pivotal role in the mechanism of melanosome transfer. On the other hand, pigmentation in the dermis is known to be caused by melanophages containing melanosomes that have dropped down from the epidermis into the dermis in abnormal conditions such as severe inflammation. Melanophages were originally described as dermal macrophages that have engulfed melanin pigment, and are also called dermal histiocytes, a representative type of phagocyte [2]. Melanosomes are thought to drop down into the dermis in cases when the basement membrane, a barrier against the invasion of foreign materials and a scaffolding for epidermal cells [3], is disrupted and loses its physiological function, which is often observed in inflammatory skin diseases such as Riehl’s melanosis, lichen planus pigmentosus and fixed drug eruptions [4,5,6].

To date, two studies have shown that dermal fibroblasts internalize melanosome clusters in areas of dermal hyperpigmentary disorders such as macular amyloidosis and also in their peripheral regions [7,8]. Further, a case report showed that dermal melanin deposition in age spots is accompanied by structurally fibroblast-like cells that had incorporated melanosome clusters [9]. More recently, many spindle-shaped dermal fibroblasts containing melanosome complexes were identified just beneath the basement membrane of chemical substance-induced leukoderma skin [10]. However, the number of reports that mention a relationship between dermal melanin pigmentation and fibroblasts is extremely limited and, to our knowledge, no studies have shown quantitative and qualitative analyses of melanosome internalization by human dermal fibroblasts. In this study, we explored the possibility that fibroblasts are involved in dermal melanin pigmentation similar to melanosome-laden dermal melanophages and may reside for a long time in the dermis.

## 2. Results

### 2.1. Melanocytes Produce Secretory Melanosome Clusters along Their Dendrites with Exposed Phosphatidylserine (PtdSer)

In order to investigate how melanocytes produce and release extracellular packages containing multiple melanosomes, termed secretory melanosome clusters, normal human melanocytes were seeded on microporous membrane filters (Figure 1a) and were cultured for 6 days. The upper and the lower surfaces of the microporous membrane filters were then observed by scanning electron microscopy (SEM) (Figure 1b–e). Since melanocyte dendrites but not their cell bodies with nuclei are able to penetrate the microporous membrane filters through the 1-μm pores [11,12], dendrites that had elongated from the melanocyte bodies were observed to be inserted into the pores of the membrane filters (Figure 1c). On the other side of the membrane, that is, on the lower surface of the membrane filter, melanocyte dendrites emerged from the membrane pores and produced several globules at various sites (Figure 1d,e). These had previously been shown to be secretory melanosome clusters [1,13,14,15,16]. The size of those clusters was larger than the pore size, indicating that they were constructed after the elongation of the dendrites through the filter. It was also observed that multiple secreted melanosome clusters released from the melanocyte dendrites had dropped down on the bottom of the culture dish (Figure 1f). The average diameter of secreted melanosome clusters measured from 500 individual clusters was 4.7 ± 1.4 µm (average ± S.D.).

Annexin V interacts with PtdSer [17], a plasma membrane phospholipid that is kept inside the cell under ordinary conditions, but is exposed on the outside of the plasma membrane to act as an ‘eat-me’ signal triggering the internalization of apoptotic cells by macrophages [18,19]. When cultured melanocytes were stained with Annexin V to detect PtdSer, fluorescence staining was observed only at some restricted sites of melanocyte dendrites (Figure 1g–i). Higher magnification revealed that the fluorescence staining of PtdSer by Annexin V was consistent with the secretory melanosome clusters generated along the various sites of dendrites (Figure 1j–l). In order to distinguish between the secretory melanosome clusters and the apoptotic bodies, we used actinomycin D, a representative apoptosis inducer, to elicit the apoptosis of melanocytes in culture following 24 h of treatment at 4 µg/mL. The results showed that the size of Annexin V/PtdSer-positive vesicles generated from actinomycin D-induced apoptotic melanocytes with shrunken nuclei was larger than the secretory melanosome clusters generated from non-apoptotic melanocytes, indicating that the secretory melanosome clusters are distinct from apoptotic bodies (Appendix A).

### 2.2. Fibroblasts Internalize Secreted Melanosome Clusters by Actin-Dependent Endocytosis

In order to evaluate the ability of fibroblasts to internalize melanosomes, isolated secreted melanosome clusters were added to the culture medium of normal human dermal fibroblasts. Time-lapse imaging at 9000-fold speed showed that fibroblasts actively moved toward and engulfed the secreted melanosome clusters (Figure 2a and Appendix A). The amount of internalized secreted melanosome clusters in fibroblasts measured by the absorbance of solubilized cells was dramatically increased in a time-dependent manner up to 48 h (Figure 2b).

Further, we investigated the possible mechanism by which the secreted melanosome clusters were internalized into fibroblasts using cytochalasin D, an inhibitor of actin polymerization. Cytochalasin D significantly decreased the internalization of secreted melanosome clusters into fibroblasts in a dose-dependent manner (Figure 2c). Although 1 µM cytochalasin D reduced cell viability by about 25%, the 2 lower concentrations of cytochalasin D tested (0.5 and 0.25 μM) had no significant cytotoxic effect (Appendix A), but elicited significant inhibitions of melanosome uptake, indicating that fibroblasts may, at least in part, internalize the secreted melanosome clusters via actin-dependent endocytosis.

### 2.3. Fibroblasts Uptake Secreted Melanosome Clusters as do Macrophages

Since dermal fibroblasts showed a high potency to internalize secreted melanosome clusters, the phagocytic ability of fibroblasts was compared to that of macrophages by adding secreted melanosome clusters to the culture medium of those cells. After 2 days of incubation, light microscopy revealed that the secreted melanosome clusters were incorporated into fibroblasts and into macrophages (Figure 3a, upper panels). Although fibroblasts and macrophages were evaluated in separate conditions, that is, in different culture media, the pellets of both types of cells were visibly pigmented (Figure 3a, lower panel). Further, the manner of internalization by each type of cell was observed by SEM (Figure 3b,c, upper panels) and by transmission electron microscopy (TEM) (Figure 3b,c, lower panels). Fibroblasts were observed to enclose secreted melanosome clusters in a membrane (Figure 3b). In contrast, macrophages, which are representative specialized phagocytes, extended their cell membranes and engulfed the secreted melanosome clusters by membrane ruffles (Figure 3c).

### 2.4. Fibroblasts Internalize Apoptotic Melanocytes

Since fibroblasts were able to internalize secreted melanosome clusters with exposed PtdSer, we assumed that apoptotic melanocytes were also phagocytosed by fibroblasts. To demonstrate this, melanocytes and fibroblasts were co-cultured in mixed medium (medium suitable for melanocytes: medium suitable for fibroblasts = 1:1) after which the culture medium was changed to medium suitable for fibroblasts or was continued in the mixed medium suitable for melanocytes and fibroblasts. After the medium was changed, the structure of melanocytes in medium suitable for fibroblasts was gradually degraded at 5 days and almost all melanocytes became fragmented and seemed to be incorporated into fibroblasts (Figure 4a). In contrast, melanocytes in the mixed medium suitable for melanocytes and fibroblasts maintained their original structure even after 10 days of culture (Figure 4b). In addition, Annexin V staining to detect PtdSer, which acts as an ‘eat-me’ signal, showed that degraded melanocytes with exposed PtdSer on the outer layer of their cell membrane were incorporated into fibroblasts, while clear-shaped and distinctive melanocytes that did not have exposed PtdSer were not incorporated into fibroblasts (Figure 4c).

Furthermore, TIM4, a receptor for PtdSer that mediates the endocytosis of phagocytes [20], was detected on fibroblasts by immunofluorescence staining (Figure 4d) and by Western blotting (Figure 4e), indicating that fibroblasts also express TIM4, which had been previously observed in macrophages [21].

## 3. Discussion

It is well known that several kinds of phagocytes exist in the dermis, such as mast cells, dendritic cells, neutrophils, monocytes and macrophages. However, the endocytic activity of dermal fibroblasts has not been studied well and has not drawn much attention, at least in the field of human dermal melanin pigmentation. A popular artificial type of dermal pigmentation is the tattoo and they are generally believed to remain essentially unchanged for the long-term. In fact, for example, the uptake and long-term storage of ink particles and latex beads by murine dermal fibroblasts has been observed [22]. In addition, melanophore-derived melanosomes are taken up by fibroblasts in frogs [23]. Further, some clinical case reports have focused on the involvement of fibroblasts in human dermal pigmentation [7,8,9,10]. Surprisingly, melanosomes were also found in cancer-associated fibroblasts before melanoma invasion in the dermis [24]. Thus, it is known that fibroblasts possess phagocytic activity, however, to our knowledge, no studies have been performed to evaluate the endocytic activity of normal human fibroblasts to internalize melanosomes. Our in vitro quantitative study of fibroblasts on the endocytosis of melanosomes combined with other basic and clinical research studies suggest that the uptake of melanogenic substances, such as secreted melanosome clusters and apoptotic melanocytes, by dermal fibroblasts may play a role in vivo as a scavenging system in the dermis.

In this study, we attempted to mimic the basement membrane using an artificial thin layer, that is, a microporous membrane filter that resembles the border between the epidermis and the dermis because there are multiple pores in the basement membrane which are conduits of immunocytes such as Langerhans cells [25]. Although we have no direct evidence of melanocyte dendrites passing through basement membrane pores, we assume that melanocyte dendrites can penetrate the pores of the basement membrane, especially when the melanosome transfer system from melanocytes to keratinocytes is disturbed. Such disruptions can impair the functional ability of keratinocytes to internalize melanosomes, such as in vitiligo [26] which is induced by the down-regulation of PAR-2, a trigger of keratinocyte ingestion [27]. Another example is melanosome cluster-laden fibroblasts that are found just below the basement membrane with no appreciable destruction of the basement membrane or hemidesmosome structure in chemical substance-induced leukoderma [10]. Taking these findings into our hypothesis, it would be feasible to think that our study using a microporous membrane filter suggests that the secreted melanosome clusters might drop down into the dermal intercellular space via the possible penetration of melanocyte dendrites and are likely to be internalized by fibroblasts that would become a kind of melanophage (Figure 5).

One of the interesting findings of our study is that fibroblasts recognize PtdSer as a landmark of substances to be internalized. PtdSer is normally located on the inner surface of the plasma membrane, but is translocated from the inner to the outer surface of the plasma membrane in apoptotic cells [18]. Among phagocytes, macrophages have been shown to recognize PtdSer exposed on the outer surface of apoptotic cells, which elicits the internalization of those cells [28]. Importantly, PtdSer is also known to be exposed on the outer surface of the plasma membrane not only when apoptosis occurs, but also in some physiological events regarding intercellular signaling such as exosomes [20]. This suggests that the expression of PtdSer on secreted melanosome clusters is a signal for recognition and phagocytosis by fibroblasts, in a similar manner as observed in ordinary exosomes. Another example is that PtdSer exposure was also observed in vesicles of membrane blebs containing a melanosome pinched off from melanocytes in chicken embryonic skin, and acts as a landmark for engulfment by neighboring keratinocytes [29]. These findings shed light on the notion that PtdSer-mediated endocytosis is utilized commonly in many events that occur in the skin.

Regarding the mechanism of endocytosis, extracellular materials are known to be internalized by two major endocytic pathways, pinocytosis and phagocytosis. Pinocytosis can be further divided into macropinocytosis, clathrin-mediated, caveolin-mediated and clathrin-/caveolae-independent endocytosis. Among these, macropinocytosis and phagocytosis are actin-dependent and both mechanisms play a role in the internalization of extracellular particles [30,31,32,33]. Although melanosomes have been shown to be internalized via multiple pathways of endocytosis [1,34,35], our data suggest that fibroblasts use an actin-dependent endocytosis pathway, maybe to potentiate the response to the internalization of various sizes of foreign invading materials. Evaluating how the amount and morphology of actin changes would be helpful to look further into the mechanism of fibroblast-mediated internalization via actin-dependent endocytosis. Surprisingly, the endocytic ability of fibroblasts to internalize secreted melanosome clusters was similar to macrophages, which suggests that fibroblasts act as one type of phagocyte in the dermis. A slight morphological difference in the phagocytic manner between fibroblasts and macrophages was observed by electron microscopy, however, further studies on the cell-specific internalization mechanisms are required.

In summary, our study demonstrates that fibroblasts actively incorporate PtdSer-exposed secretory melanosome clusters and apoptotic melanocytes in a similar manner to macrophages. Melanocyte-derived extracellular signals and melanosomes transferred to keratinocytes and other cells in the skin such as fibroblasts are thought to be able to affect the functions and metabolic status of those cells [36,37], therefore, physiological changes of fibroblasts after ingesting melanosomes remain to be determined in future research. In addition, since melanophages derived from macrophages are often found in lymph nodes of patients with inflammatory skin diseases with dermal pigmentation [38,39], it would be conceivable to hypothesize that long-lasting deposits of melanosome-laden phagocytes embedded in the dermis could be, at least in part, derived from fibroblasts. Characterizing the contribution of macrophages and fibroblasts in the removal of melanogenic substances via dermal lymph vessels would also be fascinating to evaluate in vivo in future investigations.

## 4. Materials and Methods

### 4.1. Cell Culture

Normal human melanocytes (derived from darkly pigmented newborn foreskins, passage up to 10) (Cascade Biologics, Portland, OR, USA) were maintained in medium 254 (Cascade Biologics) supplemented with a commercial cocktail of growth factors (HMGS containing phorbol 12-myristate 13-acetate (PMA), Cascade Biologics) and an antibiotic/antimycotic solution. In Figure 1a, melanocytes were seeded in transwell plates (Millipore, Billerica, MA, USA, PIRP30R48, 6-Well Millicell, Hanging Cell Culture Insert, 1.0 µm PET) at 5 × 10^5^ cells/2 mL/well using medium 254 supplemented with a commercial cocktail of growth factors (HMGS-2 containing endothelin-1, instead of HMGS containing PMA, Cascade Biologics) and were placed on the bottom of culture dishes [11], followed by incubation for 6 days with a medium change at day 3. Normal human dermal fibroblasts (derived from darkly pigmented newborn foreskins, passage up to 10) (Cascade Biologics) were cultured in medium 106 (Cascade Biologics) supplemented with a commercial cocktail of growth factors (LSGS, Cascade Biologics) and an antibiotic/antimycotic solution. In Figure 2b,c, fibroblasts were seeded in different 35-mm culture dishes at 5 × 10^5^ cells/2 mL/dish and were incubated with the indicated treatments. Macrophages used in this study were chemically differentiated cells derived from human U937 monocytic cells (DS Pharma Biomedical, Osaka, Japan). U937 cells were cultured in RPMI 1640 (DS Pharma Biomedical) supplemented with 10% fetal calf serum (HyClone, Logan, UT, USA), 2 mM glutamine and an antibiotic/antimycotic solution (Sigma-Aldrich, St Louis, MO, USA) at 37 °C in a humidified atmosphere with 5% CO_2_. In order to differentiate U937 cells to adherent phagocytes that are morphologically mature macrophage-like cells, PMA was used [40,41]. U937 cells were initially seeded in 35-mm tissue culture dishes (BD Falcon, Franklin Lakes, NJ, USA) at 1 × 10^6^ cells/2 mL/dish. Forty ng/mL PMA (Sigma-Aldrich) was continuously added to the culture medium and incubated for 7 days (at day 4, the culture medium was changed to fresh RPMI 1640 supplemented medium containing PMA and floating U937 cells were differentiated to adherent cells at this time). The medium was then further changed to RPMI 1640 supplemented medium without PMA and was incubated for 24 h before the addition of secreted melanosome clusters as shown in Figure 3.

### 4.2. PtdSer Staining of Secretory Melanosome Clusters and Apoptotic Melanocytes

The staining of PtdSer was performed using Annexin V, a calcium-dependent phospholipid-binding protein that has a high affinity for PtdSer [17]. Since Alexa Fluor^®^ 488 is resistant to the disruptive effect of melanin pigment on fluorescence intensities [42], Annexin V, Alexa Fluor^®^ 488 conjugate (Invitrogen, Carlsbad, CA, USA, A13201) was employed. Briefly, cells were washed with cold Ca^2+^ and Mg^2+^ free Dulbecco’s phosphate buffered saline (D-PBS) and then a viable cell staining solution composed of Annexin V-binding buffer (10 mM HEPES, 140 mM NaCl and 2.5 mM CaCl_2_, pH 7.4) and the same volume of culture medium (Cascade Biologics, medium 254) was added. The Annexin V, Alexa Fluor^®^ 488 conjugate was added to the viable cell staining solution at a 1:20 volume dilution. After the cells were incubated for 15 min at 37 °C in a humidified atmosphere with 5% CO_2_, the cells were observed by fluorescence microscopy (λex, 495 nm, λem, 519 nm).

### 4.3. Isolation and Treatment of Secreted Melanosome Clusters

Secreted melanosome clusters were isolated from the culture medium of normal human melanocytes derived from darkly pigmented newborn foreskins (Cascade Biologics) according to our previous method [13]. It should be noted that secreted melanosome clusters easily disappear in D-PBS, presumably due to the easy rupture of their surrounding membranes, and therefore, the culture medium should be used as a solution to collect secreted melanosome clusters. Further, the pellets of secreted melanosome clusters should be covered with a small amount of culture medium to prevent desiccation and stored at −80 °C until used. In order to incubate secreted melanosome clusters, medium appropriate to the type of cells was added to the thawed clusters, and finally, the medium containing the clusters was poured into the culture medium.

### 4.4. Treatment with an Endocytosis Inhibitor

After preliminary experiments to determine cell toxicity (Appendix A
Appendix A), final concentrations of cytochalasin D (Sigma-Aldrich), an inhibitor of actin polymerization, were employed at 0.25, 0.5 and 1 µM. Each concentration of cytochalasin D was incubated in the culture medium 30 min before the addition of secreted melanosome clusters [43]. After 2 days of incubation, the cells were washed twice with D-PBS and were harvested by trypsinization. The cells were centrifuged and the contents of internalized secretory melanosome clusters adjusted per cell number were measured spectrophotometrically [13].

### 4.5. Detection of TIM4 in Fibroblasts

TIM4, a PtdSer receptor that elicits the engulfment of apoptotic cells and PtdSer-exposed exosomes, was detected in fibroblasts by immunofluorescence staining and by Western blotting using an anti-TIM4 rabbit polyclonal antibody (Abcam, Cambridge, UK) according to standard methods. Secondary antibodies used were goat anti-rabbit IgG(H+L) labeled with Alexa Fluor^®^ 488 (Invitrogen) for immunofluorescence staining and goat anti-rabbit IgG(H+L)-HRP (Southern Biotech, Birmingham, AL, USA) for Western blotting.

### 4.6. Electron Microscopy

In order to observe secreted melanosome clusters incorporated into macrophages and fibroblasts, cells on the culture dishes were directly fixed with 2% glutaraldehyde in D-PBS at 4 °C for at least 2 h. After washing twice with D-PBS for 15 min each, the cells were post-fixed with 2% osmium tetroxide for 1.5 h. After fixation, they were dehydrated in a graded series of ethanol, and were then embedded in epoxy resin (Quetol 812) for 24 h at 60 °C. Ultrathin sections were stained with uranyl acetate and lead citrate, and were examined by TEM (JEM-1200EX, JEOL, Tokyo, Japan) at 80 kV. In order to observe melanocyte dendrite elongation and the production of secretory melanosome clusters along those dendrites, a microporous membrane filter (Millipore, 1.0 µm PET, PIRP30R48) on which melanocytes were cultured for 6 days was observed by SEM. In addition, to observe the cell surfaces of macrophages and fibroblasts by SEM, cells grown on collagen type I-coated 35 mm glass based culture dishes (IWAKI, Asahi Glass Co., Ltd., Shizuoka, Japan) were fixed with 2% glutaraldehyde in the respective culture medium at 4 °C for 2 h or more. After washing twice with D-PBS for 15 min each, the cells were post-fixed with 2% osmium tetroxide for 1.5 h. After fixation, they were dehydrated in a graded series of ethanol. Those dehydrated cells were covered with t-butyl-alcohol, freeze dried (JFD-310, JEOL, Tokyo, Japan), coated with a layer of sublimated OsO_4_ using an osmium plasma coater (OPC80, Filgen, Nagoya, Japan) and examined by SEM (JSM3620 F, JEOL) at 5 kV.

## 5. Conclusions

Our study demonstrates that dermal fibroblasts recognize PtdSer as an ‘eat-me’ signal and initiate the internalization of secreted melanosome clusters and apoptotic melanocytes, which suggests that dermal fibroblasts play a role as scavenger cells in the dermis in a manner similar to macrophages.

## Figures and Tables

**Figure 1 ijms-21-05789-f001:**
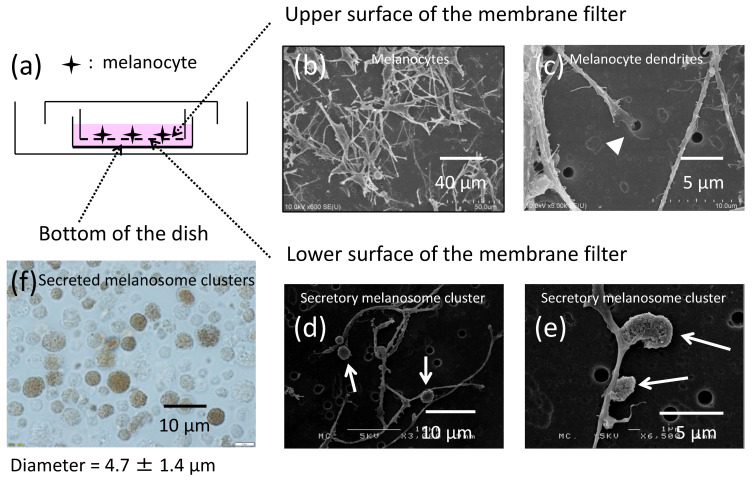
Normal human melanocytes produce secretory melanosome clusters along their dendrites with exposed PtdSer. (**a**) Normal human melanocytes were cultured for 6 days on microporous (1 µm-pore) membrane filters. Scanning electron microscopic (SEM) images of (**b**,**c**) the upper surface and (**d**,**e**) the lower surface of a membrane filter are shown. The arrow-head indicates a melanocyte dendrite inserted into a pore and the arrows indicate secretory melanosome clusters. Light microscopic bright field image of secreted melanosome clusters that had dropped down to the bottom of the culture dish released from melanocyte dendrites that had penetrated through (**f**) the microporous membrane filter. The same field of melanocytes (**g**–**i**) at lower magnification and (**j**–**l**) at higher magnification is shown with (**g**,**j**) phase contrast, (**h**,**k**) bright field and (**i**,**l**) immunofluorescence staining of PtdSer by Annexin V. Arrows indicate secretory melanosome clusters.

**Figure 2 ijms-21-05789-f002:**
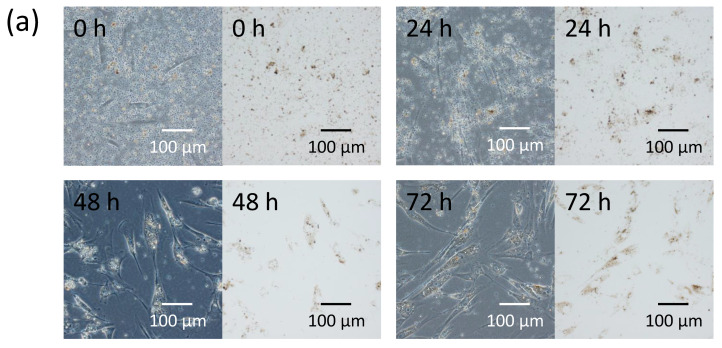
Normal human fibroblasts internalize secreted melanosome clusters in an actin-dependent manner. (**a**) Time-dependent photos from time-lapse imaging (Appendix A) and cell pellets and (**b**) melanin content per cell after the addition of secreted melanosome clusters to the culture medium of normal human fibroblasts. Cell pellets of fibroblasts pretreated for 30 min with or without an actin polymerization inhibitor, cytochalasin D (CytoD) at 0.25, 0.5 or 1 µM were produced after equal amounts of secreted melanosome clusters were added to the culture medium and were incubated for 2 days; (**c**) melanin content per cell incorporated in fibroblasts was measured spectrophotometrically. Data are expressed as a percentage of the value at 72 h (in b) or the control (ctrl) (in c) and are mean values ± SD. Statistical analysis was performed using the Dunnett II test (** *p* < 0.01 versus 24 h (in b) or the control (in c), N.S.: Not significant).

**Figure 3 ijms-21-05789-f003:**
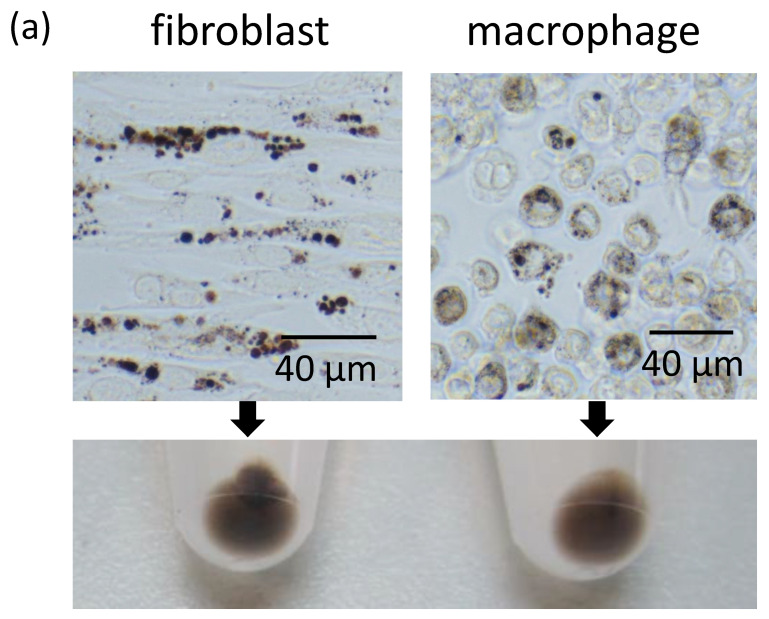
Normal human fibroblasts uptake secreted melanosome clusters as do macrophages. The same amounts of secreted melanosome clusters were added to the culture medium of fibroblasts or macrophages and were incubated for 2 days. (**a**) The upper two light microscopic images show the incorporation of secreted melanosome clusters in fibroblasts (left) and in macrophages (right). The lower photo shows the cell pellets of fibroblasts (left) and macrophages (right) collected from those cells. Sequential SEM images (upper panels) and TEM images (lower panels) of (**b**) fibroblasts and (**c**) macrophages that internalized secreted melanosome clusters are shown.

**Figure 4 ijms-21-05789-f004:**
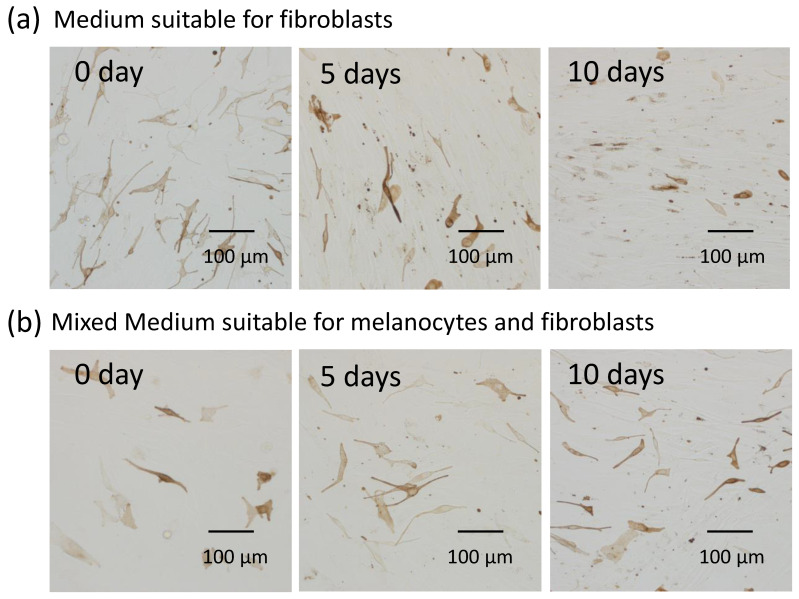
Normal human fibroblasts internalize apoptotic melanocytes and express TIM4. Time-dependent observations of melanocytes and fibroblasts co-cultured in medium suitable only for (**a**) fibroblasts or (**b**) in medium suitable for melanocytes and fibroblasts. The same fields with (left) phase contrast, (middle) bright field and (right) immunofluorescence staining of PtdSer by Annexin V showing (**c**) the manner of internalization of apoptotic melanocytes into fibroblasts. Arrows indicate apoptotic melanocytes with exposed PtdSer. Arrow-heads indicate a clear-shaped and distinctive melanocyte with no staining of Annexin V. (**d**) Immunofluorescence staining and (**e**) Western blotting of TIM4, a receptor for PtdSer, in fibroblasts is shown.

**Figure 5 ijms-21-05789-f005:**
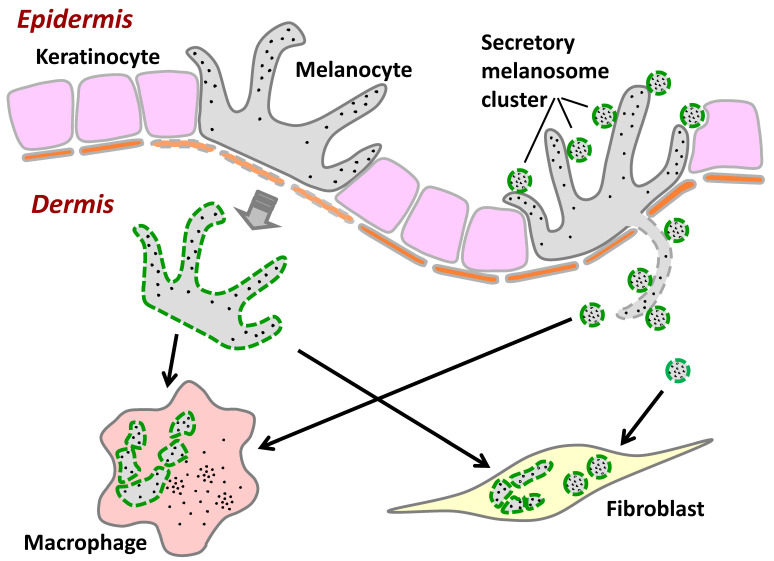
Hypothetical scheme of the internalization of secreted melanosome clusters and apoptotic melanocytes by macrophages and fibroblasts. When melanosomes or melanocytes drop down into the dermis in cases when the basement membrane is disrupted or disappears in abnormal conditions such as excessive inflammation, the secreted melanosome clusters and apoptotic melanocytes might be incorporated not only in macrophages, but also in fibroblasts. Green dashed lines indicate exposed PtdSer.

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
