# Peer review of "Dermal Fibroblasts Internalize Phosphatidylserine-Exposed Secretory Melanosome Clusters and Apoptotic Melanocytes"

_ijms, 2020, doi:10.3390/ijms21165789_

Round 1

Reviewer 1 Report

Authors performed a very interesting study with a significant possibility for  clinical application. While the experimental designs and results interpretation looks quite reasonable, reviewer has a few comments.

  • In Figure 1, microscopic observations (shown in from (g) to (l)) seem to be acquired from the upper side of membrane (same to (b) and (c)). Are the PtdSer staining also observed from the secretory melanosome clusters in bottom side of membrane? 
  • Another comments on secretory melanosome cluster formation is the potential effects of melanogenesis stimulating factors, such as UV irradiation or alpha-MSH. Did authors ever measure the such effects?
  • In line 123, page 5, the section title stating "comparable" activity between fibroblast and macrophage is anappropriate, cause it may leads to an understanding that both cells show comparable phagocytic activity. However, cause the experiments were performed in separate condition, direct comparison of phagocytic activity in both cells is impossible. 
  • Did authors observe any other kinds of PtdSer-labelled cells, other than melanocytes, that is internalized into fibroblast? 

Author Response

Reviewer #1: Comments and Suggestions for Authors (Date of this review: 02 Jun 2020)

Authors performed a very interesting study with a significant possibility for clinical application. While the experimental designs and results interpretation looks quite reasonable, reviewer has a few comments.

  • In Figure 1, microscopic observations (shown in from (g) to (l)) seem to be acquired from the upper side of membrane (same to (b) and (c)). Are the PtdSer staining also observed from the secretory melanosome clusters in bottom side of membrane?

Response – The Reviewer has brought up an interesting point but we have not performed PtdSer staining on the bottom side of the membrane. We believe that secretory melanosome clusters on the bottom side of the membrane should be stained by PtdSer, however, we thought at the time the experiment was done that PtdSer staining was enough for secretory melanosome clusters on the upper side of the membrane. Since PtdSer staining of secretory melanosome clusters on the bottom side of the membrane would be useful for the analysis of how the secretory melanosome clusters are formed, we would like to try that staining in our next study. We appreciate the Reviewer’s thoughtful idea.

  • Another comments on secretory melanosome cluster formation is the potential effects of melanogenesis stimulating factors, such as UV irradiation or alpha-MSH. Did authors ever measure the such effects?

Response – Similar to the Reviewer’s previous comment, we have not measured the effects of melanogenesis stimulating factors on the formation of secretory melanosome clusters, and we agree that such analysis could be important to better understand the role of secretory melanosome clusters. At this moment, we have an idea that replicative senescent melanocytes tend to release larger amounts of secretory melanosome clusters than young melanocytes but we have not yet determined that statistically. We really appreciate the Reviewer’s intriguing suggestion.

  • In line 123, page 5, the section title stating "comparable" activity between fibroblast and macrophage is anappropriate, cause it may leads to an understanding that both cells show comparable phagocytic activity. However, cause the experiments were performed in separate condition, direct comparison of phagocytic activity in both cells is impossible.

Response – We thank the Reviewer for this significant comment mentioning that one of our section titles was not appropriate. We agree with the Reviewer’s suggestion that the phagocytotic activities of fibroblasts and macrophages cannot be compared under different experimental conditions such as with different culture media. Therefore, in order to avoid any possible misunderstanding of the comparative evaluation of phagocytotic abilities of fibroblasts and macrophages, we made some changes and added some words about this to the revised manuscript as follows (underline).

Lines 28-30, Page 1:  The ability of dermal Dermal fibroblasts were able to uptake secreted melanosome clusters was comparable to as did macrophages, and they fibroblasts also expressed ---

Lines 132-133, Page 5:  2.3. The phagocytotic Ability of Fibroblasts Uptake Secreted Melanosome Clusters as do is Comparable to Macrophages

Lines 138-141, Page 5:  Although fibroblasts and macrophages were evaluated in separate conditions, that is, in different culture media, the pellets of bBoth types of cells were harvested and pelleted, and appeared to be visibly pigmented in a similar intensity (Figure 3a, lower panel).

Lines 149-150, Page 6:  Figure 3. Normal human fibroblasts uptake secreted melanosome clusters as do comparable to macrophages. The same amounts of sSecreted melanosome clusters ---

Line 251, Page 10:  --- melanosome clusters was comparable similar to macrophages, ---

  • Did authors observe any other kinds of PtdSer-labelled cells, other than melanocytes, that is internalized into fibroblast?

Response – Unfortunately, we have only evaluated apoptotic melanocytes being internalized into fibroblasts. When we think about PtdSer-labelled cells in the dermis, apoptotic fibroblasts as well as apoptotic macrophages are also candidates for internalization into fibroblasts. We appreciate the Reviewer’s potential idea for future investigation.

Reviewer 2 Report

The manuscript by Ando et al. provides evidence that dermal fibroblasts uptake melanosomes and apoptotic melanocytes. As the authors state in the Introduction, at least four studies have reported evidence for this phenomenon, which lowers significantly the novelty of the current manuscript. Furthermore, the authors refer several situations in which this process occurs but none of them physiological. Therefore, it is unclear what is the physiological relevance of the process described, if any, and therefore, if the term "dermal pigmentation" is relevant in non-pathological situations. Adding to this, there are several aspects related with the results presented that need to be strengthened.  

Major points:

1 - In the introduction, the authors should distinguish the process of melanosome transfer between melanocytes and keratinocytes, which is well described in physiological conditions, from the process of uptake they are now describing. Additionally, it would be appropriate to briefly mention the models of melanosome transfer and uptake by keratinocytes, since this is discussed in the Discussion section.

2 - In the Introduction, the authors should also refer to the observation that melanosomes from melanoma cells can be internalized by fibroblasts, which become CAFs [Dror, S. et al. (2016). Melanoma miRNA trafficking controls tumour primary niche formation. Nature Cell Biology, 18(9), 1006–1017. doi:10.1038/ncb3399]

3 - In Figure 1, how can the authors distinguish between Annexin V/PtdSer-positive melanosome clusters and apoptotic vesicles? Therefore, other markers of apoptotic bodies should be used and shown to be negative.

4 - Cytochalasin D is not a "representative inhibitor of phagocytosis", as it interferes with actin polymerization and therefore impairs both phagocytosis and macropinocytosis. Thus, the authors must use a specific inhibitor of phagocytosis to distinguish both processes before they can state that “fibroblasts may, at least in part, internalize the secreted melanosome clusters via actin-dependent phagocytosis” (actin-dependent phagocytosis is a redundancy).

5 - Did the authors test several doses of cytochalasin D and amiloride before deciding to use the ones presented? It is important to provide a dose-response curve.

6 - In Figure 2 (b and c), statistical analysis is required.

7 - To be able to state that "Both types of cells were harvested and pelleted, and appeared to be pigmented in a similar intensity" (lines 128 and 129), the authors need to perform quantification.

8 - What do the authors mean exactly by "cleavages of the cell membrane”? Does this imply that the membrane is disrupted? This is not clear at all from Figure 3b.  

9 - In Figure 4a, the internalization of apoptotic melanocytes is not clear. The authors should provide zoom-ins.

10 - Also in Figure 4 (d and e), the authors need to show a negative control (i.e. a cell type that does not express TIM4) and also macrophages, as a positive control. Moreover, a housekeeping gene should be used in the western-blot and the molecular weights displayed.

Minor comments:

1 - The title should be more concise, since it is too long. Also, the expression "dermal pigmentation" should be revised, since it is the epidermis that sustains skin pigmentation and, as referred, it's questionable what is the relevance, if any, of this process at the physiological level.

2 - Lines 123 and 125: "phagocytotic” should be replaced by "phagocytic".

Author Response

Reviewer #2: Comments and Suggestions for Authors (Date of this review: 10 Jun 2020)

The manuscript by Ando et al. provides evidence that dermal fibroblasts uptake melanosomes and apoptotic melanocytes. As the authors state in the Introduction, at least four studies have reported evidence for this phenomenon, which lowers significantly the novelty of the current manuscript. Furthermore, the authors refer several situations in which this process occurs but none of them physiological. Therefore, it is unclear what is the physiological relevance of the process described, if any, and therefore, if the term "dermal pigmentation" is relevant in non-pathological situations. Adding to this, there are several aspects related with the results presented that need to be strengthened.

Response – First of all, we thank the Reviewer for the dedicated efforts to improve our manuscript. As the Reviewer points out, the phenomenon that melanosomes are internalized by dermal fibroblasts is not novel because of some previous reports, however, the finding that melanocytes release PtdSer-exposed melanosome clusters which are then internalized by fibroblasts is novel. Although we have no evidence that this process occurs in the dermis of normal skin, our findings of this new function of fibroblasts could be a clue to elucidate the mechanisms of clinically observed pigmentation in the dermis.

Major points:

1 - In the introduction, the authors should distinguish the process of melanosome transfer between melanocytes and keratinocytes, which is well described in physiological conditions, from the process of uptake they are now describing. Additionally, it would be appropriate to briefly mention the models of melanosome transfer and uptake by keratinocytes, since this is discussed in the Discussion section.

Response – We appreciate the Reviewer’s suggestion to describe the melanosome transfer system in more detail. According to the Reviewer’s suggestion, a brief mention of the four models of melanosome transfer and uptake by keratinocytes was added in the “Introduction” section as follows.

Lines 43-47, Pages 1-2:  --- nuclear DNA damage. Four possible mechanisms of melanosome transfer between melanocytes and keratinocytes have been proposed, that is, the cytophagocytosis model, the membrane fusion model, the shedding-phagocytosis model and the exocytosis-endocytosis model [1]. In 3 of those models, endocytosis including phagocytosis plays a pivotal role in the mechanism of melanosome transfer. On the other hand, ---

2 - In the Introduction, the authors should also refer to the observation that melanosomes from melanoma cells can be internalized by fibroblasts, which become CAFs [Dror, S. et al. (2016). Melanoma miRNA trafficking controls tumour primary niche formation. Nature Cell Biology, 18(9), 1006–1017. doi:10.1038/ncb3399]

Response – We appreciate the Reviewer giving us the important information of another paper mentioning melanosome uptake by fibroblasts. We added that paper in the Discussion as follows.

Lines 197-198, Page 9:  --- pigmentation [7-10]. Surprisingly, melanosomes were also found in cancer-associated fibroblasts before melanoma invasion in the dermis [24]. Thus, ---

Lines 421-425, Page 14:  24.  Dror, S.; Sander, L; Schwartz, H.; Sheinboim, D.; Barzilai, A.; Dishon, Y.; Apcher, S.; Golan, T.; Greenberger, S.; Barshack, I.; Malcov, H.; Zilberberg, A.; Levin, L.; Nessling, M.; Friedmann, Y.; Igras, V.; Barzilay, O.; Vaknine, H.; Brenner, R.; Zinger, A.; Schroeder, A.; Gonen, P.; Khaled, M.; Erez, N.; Hoheisel, J.D.; Levy, C. Melanoma miRNA trafficking controls tumour primary niche formation. Nat. Cell Biol. 2016, 18, 1006-1017.

3 - In Figure 1, how can the authors distinguish between Annexin V/PtdSer-positive melanosome clusters and apoptotic vesicles? Therefore, other markers of apoptotic bodies should be used and shown to be negative.

Response – In order to demonstrate that the Annexin V-stained melanosome clusters are not fragments of apoptotic cells, both low and high magnification photos are shown in Figure 1 g-l. Since fluorescence staining of Annexin V was observed consistently in all melanosome clusters attached to melanocyte dendrites, we concluded that these clusters are PtdSer-exposed melanosome clusters. In addition, if these fluorescent clusters were fragments of apoptotic cells, it seems unlikely that all clusters would be attached to the melanocyte dendrites. Therefore, we feel that it is not necessary to perform staining with apoptosis markers other than Annexin V.

4 - Cytochalasin D is not a "representative inhibitor of phagocytosis", as it interferes with actin polymerization and therefore impairs both phagocytosis and macropinocytosis. Thus, the authors must use a specific inhibitor of phagocytosis to distinguish both processes before they can state that “fibroblasts may, at least in part, internalize the secreted melanosome clusters via actin-dependent phagocytosis” (actin-dependent phagocytosis is a redundancy).

Response – We appreciate the Reviewer’s proper and expert advice regarding the precise action of cytochalasin D. Since cytochalasin D has been used as an inhibitor of actin-related endocytosis, we changed the inaccurate words and revised the expression of the endocytosis mechanism, as follows.

Line 105, Page 4:  Fibroblasts internalize secreted melanosome clusters by actin-dependent phagocytosis actin-related endocytosis

Lines 112-118, Page 4:  Further, we investigated the possible mechanism by which the secreted melanosome clusters were internalized into fibroblasts using inhibitors of actin-dependent actin-related endocytosis; cytochalasin D, a representative an inhibitor of phagocytosis actin polymerization, and 5-(N-ethyl-N-isopropyl) amiloride (EIPA), a representative an inhibitor of macropinocytosis. Both inhibitors of actin-dependent endocytosis decreased the internalization of secreted melanosome clusters into fibroblasts (Figure 2c), indicating that fibroblasts may, at least in part, internalize the secreted melanosome clusters via actin-dependent phagocytosis actin-related endocytosis.

Lines 121-122, Page 4:  --- in an actin-dependent actin-related manner.

Line 125, Page 4:  --- 30 minutes with a phagocytosis an actin polymerization inhibitor, ---

Lines 246-247, Page 10:  --- fibroblasts use an actin-dependent phagocytosis actin-related endocytosis pathway, ---

Line 250, Page 10:  Surprisingly, the phagocytotic endocytic ability ---

5 - Did the authors test several doses of cytochalasin D and amiloride before deciding to use the ones presented? It is important to provide a dose-response curve.

Response – We determined those concentrations for evaluation after performing preliminary experiments on the reagents’ efficacy and cell toxicity at several doses. Since the purpose of this experiment was to demonstrate the inhibitory effect of cytochalasin D and amiloride (EIPA) on the internalization of secreted melanosome clusters, we did not perform experiments of dose-response curves but evaluated only the optimal concentration of each reagent. We revised the text to state that those preliminary experiments were performed and that the evaluated concentrations were determined, as follows.

Lines 312-315, Page 11:  After preliminary experiments to determine their efficacy and cell toxicity, The final concentrations of cytochalasin D (Sigma-Aldrich), a representative actin-dependent phagocytosis an inhibitor of actin polymerization, and EIPA (Sigma-Aldrich), a representative sodium/proton exchanger-dependent macropinocytosis inhibitor, were employed at 1 µM and 40 µM, respectively.

6 - In Figure 2 (b and c), statistical analysis is required.

Response – According to the Reviewer’s suggestion, we added p values for statistical analysis of the data in Figures 2b and 2c. The sentences below were added to the end of the legend for Figure 2.

Lines 129-131, Page 5:  --- spectrophotometrically (c). Data are expressed as a percentage of the value at 72 hours (in b) or the control (Ctrl) (in c) and are mean values ± SD. Statistical analysis was performed using the Dunnet II test (** p<0.01 versus 24 hours (in b) or the control (in c), N.S.: Not significant).

7 - To be able to state that "Both types of cells were harvested and pelleted, and appeared to be pigmented in a similar intensity" (lines 128 and 129), the authors need to perform quantification.

Response – Reviewer #1 also mentioned this point and we agree that it is impossible to directly compare the phagocytic abilities of fibroblasts and macrophages under different culture conditions, so we revised the text to state that fibroblasts internalize secreted melanosome clusters as do macrophages. Therefore, quantitating the absorbances of those cell pellets was not needed.

8 - What do the authors mean exactly by "cleavages of the cell membrane”? Does this imply that the membrane is disrupted? This is not clear at all from Figure 3b.

Response – We agree that the phrase “cleavages of the cell membrane” is unclear. We removed the words and modified the sentence as follows.

Lines 143-144, Page 5:  Fibroblasts were observed to internalize enclose secreted melanosome clusters through cleavages of the cell membrane and then enclosed them in a membrane (Figure 3b).

9 - In Figure 4a, the internalization of apoptotic melanocytes is not clear. The authors should provide zoom-ins.

Response – We appreciate the Reviewer’s suggestion, so we added higher magnification photos in Figure 4a and 4b. At the same time, we removed the phase contrast images in order to save space. Now the internalization of apoptotic melanocytes is now clear.

10 - Also in Figure 4 (d and e), the authors need to show a negative control (i.e. a cell type that does not express TIM4) and also macrophages, as a positive control. Moreover, a housekeeping gene should be used in the western-blot and the molecular weights displayed.

Response – Actually, the negative control staining was obtained by adding only the secondary antibody without the primary TIM4 antibody but we omitted it in the original manuscript. Since the validity of TIM4 fluorescent staining can be guaranteed by the photo of negative control 1st antibody (-) & 2nd antibody (+), we think that it is not necessary to stain cells that do not express TIM4. A paper providing the specificity of TIM4 for immunohistochemical staining using the same Abcam TIM4 antibody (ab47637) was published by “Zhang Q et al. TIM-4 promotes the growth of non-small-cell lung cancer in a RGD motif-dependent manner. Br. J. Cancer 2015, 113, 1484-1492”. In addition, TIM4 is known to be expressed in macrophages (a paper providing that has been added in the revised manuscript as reference #21), so we do not think that is necessary either. Moreover, we already had an image of the housekeeping gene (β-actin) re-blotted on the TIM4 Western-blot membrane, so we added that to the Figure according to the Reviewer’s suggestion, together with molecular weight markers of TIM4.

Lines 173-174:  ---which had been previously observed in macrophages [21].

Lines 414-415, Page 14:  21.  Nishi, C.; Toda, S.; Segawa, K.; Nagata, S. Tim4- and MerTK-mediated engulfment of apoptotic cells by mouse resident peritoneal macrophages. Mol. Cell. Biol. 2014, 34, 1512-1520.

Minor comments:

1 - The title should be more concise, since it is too long. Also, the expression "dermal pigmentation" should be revised, since it is the epidermis that sustains skin pigmentation and, as referred, it's questionable what is the relevance, if any, of this process at the physiological level.

Response – We agree with the Reviewer’s suggestion that the phrase “dermal pigmentation” is confusing. In order to focus on cellular physiology, we removed the latter part of the title as follows.

Lines 4-5, Page 1:  Dermal fibroblasts internalize phosphatidylserine-exposed secretory melanosome clusters and apoptotic melanocytes: Possible involvement of fibroblasts in dermal pigmentation

2 - Lines 123 and 125: "phagocytotic” should be replaced by "phagocytic".

Response – According to the Reviewer’s suggestion, we changed the words “phagocytotic” and “endocytotic” to “phagocytic” and “endocytic”, respectively. The section title has been changed and now there is no “phagocytotic”.

Lines 134-135, Page 5:  --- the phagocytotic phagocytic ability of fibroblasts ---

Line 190, Page 8:  --- However, the endocytotic endocytic activity of ---

Line 199, Page 9:  --- possess phagocytotic phagocytic activity, ---

Line 200, Page 9:  --- evaluate the endocytotic endocytic activity ---

Line 241, Page 10:  --- two major endocytotic endocytic pathways, ---

Line 250, Page 10:  Surprisingly, the phagocytotic endocytic ability of ---

Lines 252-253, Page 10:  --- difference in the phagocytotic phagocytic manner ---

Reviewer 3 Report

This study is very interesting and the manuscript is well written.

It would be better if several explanation is added in the manuscript.

  1. When fibroblast internalize the sereted melanosome via actin-dependent phagocytosis, how is the amount or morphology of actin protein changed ? It would be better if the authors observe or discuss it.
  2. After fibroblasts eat melanosome or melanocyte debris, how do fibroblasts deal with it? Is there no effect on the survival of fibroblasts ?

Author Response

Reviewer #3: Comments and Suggestions for Authors (Date of this review: 05 Jun 2020)

This study is very interesting and the manuscript is well written.

It would be better if several explanation is added in the manuscript.

  1. When fibroblast internalize the sereted melanosome via actin-dependent phagocytosis, how is the amount or morphology of actin protein changed ? It would be better if the authors observe or discuss it.

Response – The Reviewer’s suggestion is very interesting and important but we have not yet performed such experiments. According to this suggestion, we added a sentence in the text as follows.

Lines 247-249, Page 10:  --- foreign invading materials. Evaluating how the amount and morphology of actin changes would be helpful to look further into the mechanism of fibroblast-mediated internalization via actin-related endocytosis. Surprisingly, ---

  1. After fibroblasts eat melanosome or melanocyte debris, how do fibroblasts deal with it? Is there no effect on the survival of fibroblasts ?

Response – We appreciate the Reviewer’s fascinating question about the survival and responses of fibroblasts after their internalization of melanogenic substances. We have not yet evaluated such effects, but we plan to investigate the physiological changes of fibroblasts after their internalization of melanogenic substances in the future and will report that as an independent study.

Round 2

Reviewer 2 Report

The authors revised the manuscript but do not provide any new data to answer some of the Reviewer's major points. Moreover, the authors preferred to downplay several of their claims, significantly reducing the relevance and impact of their work. For example, the authors chose to claim that the internalization of melanosomes by fibroblasts is actin-dependent. If this is the case, why using an inhibitor of macropinocytosis, an actin-dependent process? More importantly, it is unlikely that any endocytosis route is actin-independent, so this claim lacks any relevance if not accompanied by more specific ways of defining the internalization route. Therefore, the authors must at least address the following points by providing new data:

1 - Use a marker of apoptotic bodies to distinguish these form Annexin V/PtdSer-positive melanosome clusters.

2 - Use an inhibitor of phagocytosis to assess if this internalization route is involved.

The authors should also provide the data that led them to conclude that the doses of the drugs used are efficient and do not cause cell toxicity, in supplementary info.

Minor points:

1 - In the Introduction, the words "endocytosis including" should be deleted from the new sentence describing the transfer mechanisms.

2 - "actin-related endocytosis" should be replaced by "actin-dependent endocytosis".

3 - The control for the TIM4 antibody is not required to be shown in the case of the immunofluorescence.

Author Response

July 17, 2020

Ling Zhu

Assistant Editor, IJMS Editorial Office

Dear Ms. Ling Zhu,

                 Thank you for sending us the further comments of Reviewer #2 regarding our revised manuscript (ijms-833817) resubmitted to the International Journal of Molecular Sciences, which invited us to further revise and resubmit this manuscript. We have performed new experiments as requested and have included those results and made a number of other revisions in the manuscript according to the comments and suggestions of Reviewer #2. We are pleased to now resubmit the revised manuscript. We thank you and the Reviewers for the time and effort to critique and thus improve this manuscript and we feel that the revised version is a stronger article as a result.

                 The name of the person who performed the new experiment (Ms. Nene Shimoda) has been added as an author.

                 We believe that we have answered each of the critiques raised by Reviewer #2, as detailed point by point on the following pages and we have marked modified text in the revised manuscript with Track Changes. We hope that these new modifications will satisfy Reviewer #2 and that the manuscript can now be accepted for publication in the International Journal of Molecular Sciences. We look forward to hearing from you and please let me know if you require anything further regarding this revision.

Sincerely,

Hideya Ando

*************************************************

Prof. Hideya Ando, Ph.D.

Department of Applied Chemistry and Biotechnology

Okayama University of Science

1-1 Ridai-cho, Kita-ku, Okayama 700-0005, Japan

Tel&Fax +81-86-256-9726 / E-mail: [email protected]

*************************************************

Responses to the Reviewers' comments:

Reviewer #2: Comments and Suggestions for Authors
-Round 2 (Date of this review: 25 Jun 2020)

The authors revised the manuscript but do not provide any new data to answer some of the Reviewer's major points. Moreover, the authors preferred to downplay several of their claims, significantly reducing the relevance and impact of their work. For example, the authors chose to claim that the internalization of melanosomes by fibroblasts is actin-dependent. If this is the case, why using an inhibitor of macropinocytosis, an actin-dependent process? More importantly, it is unlikely that any endocytosis route is actin-independent, so this claim lacks any relevance if not accompanied by more specific ways of defining the internalization route. Therefore, the authors must at least address the following points by providing new data:

Response – Thank you very much for your important comments and suggestions to further improve our manuscript. As we mention in the manuscript (Lines 249-253), our understanding of endocytosis is that phagocytosis and macropinocytosis require actin polymerization, that is, those two endocytic pathways are actin-dependent events, while other types of endocytosis, such as clathrin-/caveolin-mediated pinocytosis, are actin-independent. In this manuscript regarding internalization, we just wanted to propose that fibroblasts internalize secreted melanosome clusters via actin-dependent endocytosis, but did not intend to distinguish between the processes of phagocytosis and macropinocytosis. Therefore, in order to clarify our message taking the Reviewer’s comments into account, we modified the manuscript, as follows.

Lines 120-127, Page 4:  Further, we investigated the possible mechanism by which the secreted melanosome clusters were internalized into fibroblasts using inhibitors of actin-dependent endocytosis; cytochalasin D, a representative an inhibitor of phagocytosis actin polymerization, and 5-(N-ethyl-N-isopropyl) amiloride (EIPA), a representative an inhibitor of Na+/H+ exchange at the cell surface that leads to the inhibition of actin-dependent macropinocytosis [20]. Both of those inhibitors of actin dependent endocytosis significantly decreased the internalization of secreted melanosome clusters into fibroblasts (Figure 2c), indicating that fibroblasts may, at least in part, internalize the secreted melanosome clusters via actin-dependent phagocytosis endocytosis.

Lines 431-432, Page 14:  20. Lagana, A., Vadnais, J., Le, P.U., Nguyen, T.N., Laprade, R., Nabi, I.R., Noël, J. Regulation of the formation of tumor cell pseudopodia by the Na+/H+ exchanger NHE1. J. Cell Sci. 2000, 113, 3649-3662.

1 - Use a marker of apoptotic bodies to distinguish these form Annexin V/PtdSer-positive melanosome clusters.

Response – The Reviewer mentioned in a previous comment (in Round 1) that “how can the authors distinguish between Annexin V/PtdSer-positive melanosome clusters and apoptotic vesicles? Therefore, other markers of apoptotic bodies should be used and shown to be negative.” Since apoptotic bodies also have exposed PtdSer, it is not easy to discriminate between apoptotic bodies and exosome-like secretory melanosome clusters using a staining method. Therefore, in order to investigate the differences between apoptotic bodies and secretory melanosome clusters, we employed actinomycin D, a representative apoptosis inducer, to elicit apoptosis in cultured normal human melanocytes (24 hours incubation at 4 μg/ml) and then observed the difference between secretory melanosome clusters and apoptotic bodies. The results showed that the size of apoptotic bodies generated from actinomycin D-induced apoptotic melanocytes with shrunken nuclei were larger than the secretory melanosome clusters generated from non-apoptotic melanocytes. Furthermore, non-apoptotic melanocytes did not produce such a large size of vesicles but produced only small monotonous sized vesicles. These results indicate that the secretory melanosome clusters are distinct from apoptotic bodies. Those images are shown in Supplementary Materials as Figure S1, as follows.

Lines 92-99, Pages 2-3:  --- along the various sites of dendrites (Figure 1j-l). In order to distinguish between the secretory melanosome clusters and the apoptotic bodies, we used actinomycin D, a representative apoptosis inducer, to elicit the apoptosis of melanocytes in culture following 24 hours of treatment at 4 μg/ml. The results showed that the size of Annexin V/PtdSer-positive vesicles generated from actinomycin D-induced apoptotic melanocytes with shrunken nuclei was larger than the secretory melanosome clusters generated from non-apoptotic melanocytes, indicating that the secretory melanosome clusters are distinct from apoptotic bodies (Supplementary Material Figure S1).

Lines 360-364, Page 13:  Supplementary Materials: Figure S1. The same fields of non-apoptotic melanocytes (a) and actinomycin D-induced apoptotic melanocytes (24 hours incubation at 4 μg/ml) (b) are shown with bright field (left panels), DNA staining by Hoechst 33342 (middle panels) and immunofluorescence staining of PtdSer by Annexin V (right panels). Asterisks in (a) indicate secretory melanosome clusters. Arrows in (b) indicate apoptotic bodies without nuclear DNA. Arrow-heads in (b) indicate apoptotic melanocytes with shrunken nuclei and PtdSer exposure on the cell membrane.

2 - Use an inhibitor of phagocytosis to assess if this internalization route is involved.

Response – As we mention above, we used inhibitors such as cytochalasin D and EIPA to demonstrate that fibroblasts may, at least in part, internalize the secreted melanosome clusters via actin-dependent endocytosis. We believe that those inhibitors, cytochalasin D and EIPA, are suitable to prove our claim. If the Reviewer could provide information about an inhibitor of phagocytosis that would be more definitive, it would be very much appreciated.

The authors should also provide the data that led them to conclude that the doses of the drugs used are efficient and do not cause cell toxicity, in supplementary info.

Response – We performed MTS assays to evaluate the effects of cytochalasin D and EIPA on cell viability, respectively. Since the evaluated concentrations of cytochalasin D (1 μM) and EIPA (40 μM) used in this study slightly decreased cell viability, the contents of internalized secreted melanosome clusters in fibroblasts were adjusted per cell number and the inhibitory effects of internalization were observed statistically. The results are shown in Supplementary Materials as Figure S2, as follows.

Lines 321-322, Page 12:  After preliminary experiments to determine their efficacy and cell toxicity (Supplementary Materials Figure S2), The final concentrations of cytochalasin D ---

Lines 327-329, Page 12:  --- The cells were centrifuged and melanin contents in the pellets the contents of internalized secretory melanosome clusters adjusted per cell number were measured spectrophotometrically [1213].

Lines 366-367, Page 13:  Figure S2. Cell viability determined using the MTS assay after treatment with or without various concentrations of cytochalasin D (a) or EIPA (b) for 24 hours.

Minor points:

1 - In the Introduction, the words "endocytosis including" should be deleted from the new sentence describing the transfer mechanisms.

Response – According to the Reviewer’s suggestion, we deleted the words “endocytosis including” from the new sentence in the Introduction as follows:

Line 46, Page 2:  In 3 of those models, endocytosis including phagocytosis ---

2 - "actin-related endocytosis" should be replaced by "actin-dependent endocytosis".

Response – According to the Reviewer’s suggestion, we replaced “actin-related endocytosis” by “actin-dependent endocytosis” as follows:

Lines 112-113, Page 4:  Fibroblasts internalize secreted melanosome clusters by actin-dependent phagocytosis actin-related dependent endocytosis

Line 127, Page 4:  --- melanosome clusters via actin-dependent phagocytosis actin-related dependent endocytosis.

Lines 255-256, Pages 10-11:  --- fibroblasts use an actin-dependent phagocytosis actin-related dependent endocytosis pathway, ---

3 - The control for the TIM4 antibody is not required to be shown in the case of the immunofluorescence.

Response – According to the reviewer’s suggestion, we removed the photo of negative control, 1st antibody (-) & 2nd antibody (+), from Fig. 4d.

Round 3

Reviewer 2 Report

The authors provide new data regarding the distinction between apoptotic bodies and melanosome clusters and the toxicity of the drug doses used. In the case of the latter, the authors now show around 25% reduction in cell viability with the dose of cytochalasin D used. This is not a slight decrease, as mentioned in the previous rebuttal letter, and must be mentioned in the manuscript.

The other issue remaining is the conclusion taken about the internalization route. As the authors "just wanted to propose that fibroblasts internalize secreted melanosome clusters via actin-dependent endocytosis", they should only show the results from cytochalasin D and omit all the experiments and references to the use of EIPA. If the authors want to use the EIPA to discriminate between macropinocytosis and phagocytosis, they must use other strategies, such as the depletion of Rac/Rho family proteins.

Author Response

August 3, 2020

Icey Wang

Section Managing Editor, IJMS Editorial Office

Dear Ms. Icey Wang,

                 Thank you for sending us the Round 3 comments of Reviewer #2 regarding our revised manuscript (ijms-833817) resubmitted to the International Journal of Molecular Sciences, which invited us to further revise and resubmit this manuscript. Taking the Reviewer’s comments and suggestions into account, we have performed an additional experiment and we are pleased to resubmit the revised manuscript. We thank you and the Reviewers for the time and effort to critique and thus improve this manuscript and we feel that the revised version is a stronger article as a result.

                 We believe that we have answered all the critiques raised by Reviewer #2, as detailed point by point on the following pages and we have marked modified text in the revised manuscript with Track Changes. We hope that these new modifications will satisfy Reviewer #2 and that the manuscript can now be accepted for publication in the International Journal of Molecular Sciences. We look forward to hearing from you and please let me know if you require anything further regarding this revision.

Sincerely,

Hideya Ando

*************************************************

Prof. Hideya Ando, Ph.D.

Department of Applied Chemistry and Biotechnology

Okayama University of Science

1-1 Ridai-cho, Kita-ku, Okayama 700-0005, Japan

Tel&Fax +81-86-256-9726 / E-mail: [email protected]

*************************************************

Responses to the Reviewers' comments:

Reviewer #2: Comments and Suggestions for Authors
-Round 3 (Date of this review: 24 July 2020)

The authors provide new data regarding the distinction between apoptotic bodies and melanosome clusters and the toxicity of the drug doses used. In the case of the latter, the authors now show around 25% reduction in cell viability with the dose of cytochalasin D used. This is not a slight decrease, as mentioned in the previous rebuttal letter, and must be mentioned in the manuscript.

The other issue remaining is the conclusion taken about the internalization route. As the authors "just wanted to propose that fibroblasts internalize secreted melanosome clusters via actin-dependent endocytosis", they should only show the results from cytochalasin D and omit all the experiments and references to the use of EIPA. If the authors want to use the EIPA to discriminate between macropinocytosis and phagocytosis, they must use other strategies, such as the depletion of Rac/Rho family proteins.

Response – Thank you very much for these valuable comments and information.

  • According to your suggestion, we omitted all experiments and references regarding EIPA.

Lines 118-120, Page 4:  ---using cytochalasin D, an inhibitor of actin polymerization. , and 5-(N-ethyl-N-isopropyl) amiloride (EIPA), a representative an inhibitor of Na+/H+ exchange at the cell surface that leads to the inhibition of actin-dependent macropinocytosis [20].

Lines 133-135, Page 5:  --- 30 minutes with or without an actin polymerization inhibitor, cytochalasin D (CytoD), or a macropinocytosis inhibitor, EIPA, at 0.25, 0.5 or 1 µM and 40 µM, respectively, were produced ---

Lines 316-321, Page 11:  After preliminary experiments to determine their efficacy and cell toxicity (Supplementary Materials Figure S2), final concentrations of cytochalasin D (Sigma-Aldrich), an inhibitor of actin polymerization, and EIPA (Sigma-Aldrich), a sodium/proton exchanger-dependent macropinocytosis inhibitor, were were employed at 0.25, 0.5 and 1 µM. and 40 μM, respectively. Those inhibitors were Each concentration of cytochalasin D was incubated in the culture medium 30 minutes before the addition of secreted melanosome clusters [4441].

Page 13:  (Remove the reference #20 regarding EIPA)

  1. Lagana, A., Vadnais, J., Le, P.U., Nguyen, T.N., Laprade, R., Nabi, I.R., Noël, J. Regulation of the formation of tumor cell pseudopodia by the Na+/H+ exchanger NHE1. J. Cell Sci. 2000, 113, 3649-3662.

Page 14:  (Remove the references regarding macropinocytosis)

  1. Kerr, M.C.; Teasdale, R.D. Defining macropinocytosis. Traffic 2009, 10, 364-371.
  2. Mercer, J.; Helenius, A. Virus entry by macropinocytosis. Nat. Cell Biol. 2009, 11, 510-520.

Supplementary Materials, Figure S2:  (Remove the data of EIPA)

Figure S2. Cell viability determined using the MTS assay after treatment with or without various concentrations of cytochalasin D (a) or EIPA (b) for 24 hours. Data represent means ± SD; **=p<0.01 versus 0 μM; N.S.=Not significant.

  • As for the 25% reduction in cell viability with the dose of cytochalasin D used (1 μM), we agreed with your comment mentioning that “This is not a slight decrease”. Taking the previous comment (Round 1) of Reviewer #2 that we should “provide a dose-response curve” into account, we performed an additional experiment determining the dose-dependent efficacy of cytochalasin D at 0.25, 0.5 and 1 μM, the 2 lower concentrations having no significant effect on cytotoxicity (Figure S2). The data in original Figure 2c were replaced by those data, which indicate that cytochalasin D has a potency to inhibit the internalization of secreted melanosome clusters by fibroblasts at lower concentrations that do not significantly affect cell viability.

Lines 120-124, Page 4:  Both of those inhibitors Cytochalasin D significantly decreased the internalization of secreted melanosome clusters into fibroblasts in a dose-dependent manner (Figure 2c). Although 1 μM cytochalasin D reduced cell viability by about 25%, the 2 lower concentrations of cytochalasin D tested (0.5 and 0.25 μM) had no significant cytotoxic effect (Figure S2) but elicited significant inhibitions of melanosome uptake, indicating that ---

Lines 127-129, Pages 4-5:  (Figure 2c was replaced.)

  • In addition, we added an explanation of Figure 4c, as follows.

Lines 186-189, Page 8:  The same fields with phase contrast (left), bright field (middle) and immunofluorescence staining of PtdSer by Annexin V (right) are shown showing the manner of internalization of apoptotic melanocytes into fibroblasts (c).
